# Cardiac Amyloidosis Therapy: A Systematic Review

Franco Iodice, Marco Di Mauro ⬤, Marco Giuseppe Migliaccio, Angela Iannuzzi, Roberta Pacileo, Martina Caiazza and Augusto Esposito *⬤

Department of Translational Medical Sciences, University of Campania "Luigi Vanvitelli", 81100 Naples, Italy; francoiodice@yahoo.it (F.I.); marco.dimauro4@gmail.com (M.D.M.); marcog.migliaccio@gmail.com (M.G.M.); ang.iannuzzi@gmail.com (A.I.); pacileoroberta@gmail.com (R.P.); martina.caiazza@yahoo.it (M.C.)
* Correspondence: augustoesposito1990@gmail.com

**Abstract:** Heart involvement in Cardiac Amyloidosis (CA) results in a worsening of the prognosis in almost all patients with both light-chain (AL) and transthyretin amyloidosis (ATTR). The mainstream CA is a restrictive cardiomyopathy with hypertrophic phenotype at cardiac imaging that clinically leads to heart failure with preserved ejection fraction (HFpEF). An early diagnosis is essential to reduce cardiac damage and to improve the prognosis. Many therapies are available, but most of them have late benefits to cardiac function; for this reason, novel therapies are going to come soon.

**Keywords:** cardiac amyloidosis; therapies amyloidosis





## 1. Introduction

Amyloidosis is a rare condition caused by an abnormal extracellular deposition of a misfolded protein, called amyloid, potentially affecting any organs and tissues.

Among the different types of this clinical disorder, more than 95% of all cardiac amyloidosis (CA) are associated with immunoglobulin light-chain amyloidosis (AL) and transthyretin amyloidosis (ATTR) [1].

AL, previously known as primary amyloidosis, is related to the misfolding of antibody light chain fragments produced by a plasma-cell clone. The estimated incidence is 3–12 cases per million persons per year, with a prevalence of 30,000 to 45,000 affected in the United States and European Union [2].

ATTR is related to the abnormal production of a type of amyloid called transthyretin (TTR, previously called prealbumin), which is primarily made by the liver, and it can be either acquired or congenital. The acquired wild-type variant (ATTRwt) typically affects older males, and is therefore named "senile amyloidosis". The mutant variant (ATTRm or Hattr—hereditary) derives from mutations in the TTR gene and it may present mostly with cardiac symptoms, neurological symptoms, or both.

Chronic amyloid fibril deposition in myocardium leads to the development of hypertrophic cardiomyopathy and diastolic dysfunction. Dyspnea on exertion and atrial fibrillation with cardioembolic events are the most frequent onset symptoms. Low extremity edema and ascites may also occur in patients with prevalent right-heart involvement [3]. Other findings are bundle branch block and complete atrioventricular block. A deep evaluation of clinical symptoms, cardiac and systemic involvement (biomarkers and cardiac imaging, serum, urine testing, biopsy) can help to diagnose amyloidosis and to differentiate it into AL or ATTR.

Treatment of CA aims to improve symptoms and limit further production of amyloid protein. Specific treatments vary depending on the type of amyloidosis (see Figures 1 and 2) [4].

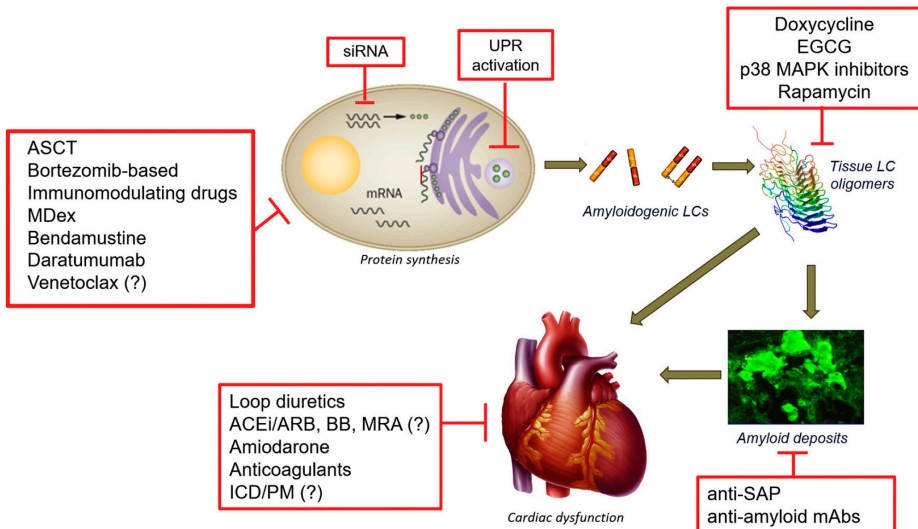

**Figure 1.** Targets for treatment in cardiac light-chain (AL) amyloidosis [4].

| AL | | | ATTR | | |
|---|---|---|---|---|---|
| **Anti-plasma cell therapies** | **Alkylating agents** | Melphalan | **TTR silencers** | siRNA | Patisiran |
| | | Cyclophosphamide | | ASO | Inotersen (IONI-TTR Rx) |
| | **Proteasome inhibitors** | Bortezomib | **TTR stabilizers** | Diflunisal | |
| | | Ixazomib | | Tafamidis | |
| | **Immunomodulators** | Pomalidomide | | Tolcapone | |
| | **Anti-CD38 monoclonal antibody** | Daratumumab | | AG10 | |
| **Anti-amyloid antibody** | NEOD001 | | **Fibril disruptors** | Doxycycline + TUDCA | |
| | | | | Green tea extract | |
| | | | | Curcumin | |
| | | | | Anti-amyloid antibody | PRX004 |
| **Ubiquitous Anti-Amyloid Fibril Antibody** | | | | | |
| Monoclonal IgG1 anti-SAP antibody | | | | | |

**Figure 2.** Amyloid-specific pharmacotherapies.

## 2. Supportive Care

Conventional therapy of heart failure, such as ACEi/ARB, diuretics and beta-blockers, may be poorly tolerated [5,6]. Poor ventricular compliance implicates higher filling pressure to reach a pre-load fitting. Diuretic therapy needs attention because of the impairment of preload, which can reduce systolic output and cause cerebral hypoperfusion. Furthermore, mostly in cases of restrictive pathophysiology, cardiac output is strictly related to heart rate. In addition to beta-blockers, digoxin is an option, but it should be used with caution, because of the binding to the amyloid fibrils and the consequent rise in their levels [7–9].

Progressive deterioration of ejection fraction and heart failure, but also bradyarrhythmia and electrical mechanical dissociation, can increase sudden cardiac death (SCD) risk [10]. A prophylactic placement of an implantable cardioverter-defibrillator (ICD)

should be considered when syncope and complex non-sustained ventricular arrhythmias occur [11].

Subcutaneous devices (s-ICD) can reduce the defibrillation threshold and result in successful defibrillation [12]. More trials are needed to better understand and recognize predictors of arrhythmia associated with SCD in CA.

Early insertion of a pacemaker may help to treat symptomatic bradyarrhythmias. In familial amyloidosis, selected electrophysiological criteria have been published to guide pacemaker placement: His-ventricular interval $\geq$ 70 ms, His-ventricular interval > 55 ms (if associated to fascicular block), Wenckebach anterograde point $\leq$ 100 beats/min. In fact, in these conditions, there is a high-degree atrioventricular block risk [13].

### 2.1. Light Chain Amyloidosis (AL)

Without treatment, AL amyloidosis has a progressive course due to uncontrolled organ damage. Thus, patients with systemic disease should immediately initiate therapy, and there are now multiple novel therapeutics approved or under development [14–16].

### 2.1.1. Classical Therapies Targeting the Plasma Cells

For primary amyloidosis, treatments include the same agents used to treat multiple myeloma (MM), with adaptations for dosages and schedule [17]. In patients with good cardiac, liver and renal function, the first-line therapy is usually a combination of high-dose melphalan (HDM) and autologous stem cell transplantation (ASCT). An induction therapy before HDM/ASCT can be assessed with Cyclophosphamide, Vincristine and Dacarbazine (CVD) combination treatment.

If ASCT is not a suitable option for the patient, an oral Melphalan and Dexamethasone (MDex)-based regimen is considered an effective and well-tolerated therapeutic scheme [18].

A milestone for the treatment has been represented by the introduction of proteasome inhibitors. Bortezomib is a reversible inhibitor of the 26S proteasome involved in the degradation of ubiquitous proteins. It can rapidly reduce misfolded light chains' concentration and it represents a standard of care in non-transplant-eligible patients [19].

Ixazomib is a second-generation oral proteasome inhibitor authorized for the treatment of MM combined with Lenalidomide and Desametasone. Several randomized studies are currently ongoing, comparing Ixazomib with standard of care.

Effectiveness and safety of Carfilzomib has been evaluated in a phase I/II study, in patients with relapsed/refractory AL amyloidosis, showing an acceptable response rate, despite an increased cardiopulmonary toxicity [20].

A good clinical efficacy has been shown for relapsed or refractory disease by Thalidomide, Lenalidomide and Pomalidomide, even if their mechanism of action is not completely understood [21] and the toxicity profile is not insignificant. Thalidomide is more effective when used in association, but it causes neurological toxicity and it is teratogenic. Lenalidomide is more tolerated, although it needs renal function monitoring. Pomalidomide is a novel immunomodulating drug for the AL amyloidosis and is now under study in a phase 2 trial.

### 2.1.2. Alternative Strategies Targeting the Plasma Cell

Increasing research into new therapeutic strategies is essential, since most patients do not reach a complete hematological response to the standard therapy.

Alkylating agents causing DNA damages, such as Bendamustine, are used for the treatment of MM and Waldenström macroglobulinemia [22]. A survival benefit and an efficient hematological response have been proven in a recent study evaluating Bendamustine and Prednisone association [23]. The safety and efficacy of Bendamustine and dexamethasone are under evaluation in patients with relapsed AL amyloidosis and advanced disease stages.

New therapies for relapsed/refractory MM in the advanced cardiac AL amyloidosis include Daratumumab, a human IgG1k monoclonal antibody targeting the CD38 antigen on plasma cells (also expressed in AL amyloidosis) [24,25]. Venetoclax is a BH3-mimetic used for chronic lymphocytic leukemia (CLL), and has a proven role in patients with translocation t(11;14), most commonly associated with AL amyloidosis.

### 2.1.3. Therapeutic Strategies Targeting Light Chain Aggregation

New therapeutic strategies targeting light-chain (LC) aggregation aims to prevent their cardiotoxic effects and prevent the further formation of amyloid fibrils.

Doxycycline seems to play a role on both fronts [26]. In a phase II open label trial (DUAL trial), oral doxycycline is administered together with plasma cell-directed drugs [27]. Bortezomib, Cyclophosphamide and Dexamethasone are being evaluated in association with Doxycycline in a randomized phase II/III trial [28].

Epigallocatechin-3-gallate (EGCG), a polyphenol able to convert amyloid fibers into unstructured oligomers, is proven to prevent amyloid formation in vitro [29]. Extracellular LC deposition causes contractile dysfunction by promoting apoptosis through a mitogen-activated protein kinase (MAPK) pathway. In this regard, SB203580, a selective p38 MAPK inhibitor, has been shown to reduce oxidative stress and apoptosis induced by extracellular LC in AL amyloidosis-cultured cardiomyocytes [30]. Moreover, LC-mediated cardiotoxicity seems to be sustained by a dysregulation of the autophagic pathway, which could be restored by Rapamycin (Sirolimus), but the modulation of this pathway could not be considered for new therapies to date.

### 2.1.4. Therapeutic Strategies Targeting Amyloid Deposits

Systemic amyloid deposits are typically not surrounded by an inflammatory reaction [31], and their rapid elimination is necessary to restore vital organ function. Therefore, new therapeutic strategies have the aim to trigger an immune response against amyloid fibrils. Thus, the potential use and efficacy of a serum amyloid P component (SAP) binding molecule, R-(1-[6-[(R)-2-carboxy-pyrrolidin-1-yl]-6-oxo-exanoyl]-pyrroli-dine-2-carboxylicacid (CPHPC) [32] will be clarified in further studies.

Passive immunotherapy is an additional promising pharmaceutical strategy using specific antibodies targeting the misfolded LC proteins.

A humanized murine monoclonal antibody, NEOD001, has been shown to react with LC aggregates, thus increasing AL amyloid clearance in a murine model [33], but two phase II/III trials have been interrupted due to negative results.

## 3. TTR Amyloidosis

Many new therapies for TTR amyloidosis have been developed in recent years, in addition to conventional supportive therapy.

### 3.1. Liver Transplantation

Liver transplantation (LTx) was the conventional first-line therapy for patients with mutant ATTR [34] since 1990. Data contained in The Familial Amyloidotic Polyneuropathy World Transplant Registry (FAPWTR) show an excellent long-term survival in well-selected patients, underlying that an optimal nutritional status, an early onset and a short duration of disease at the time of LTx are independent factors for survival. Furthermore, non-TTR Val30Met patients have a worse outcome compared to TTR Val30Met patients (Figure 3), as demonstrated by the continuing production of fibrils derived from wild- type TTR after LTx in some autopsy studies [35].

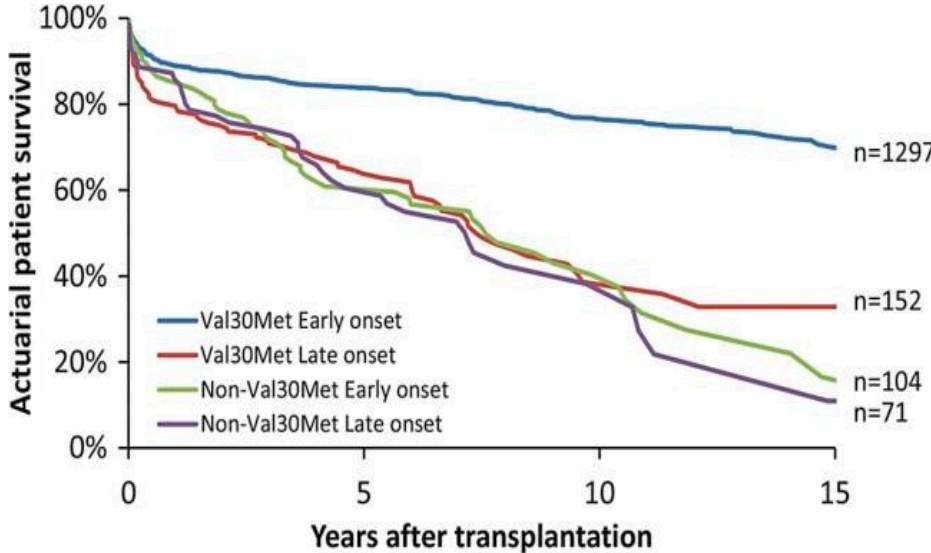

**Figure 3.** Patient survival between 1990 and 2010. Comparison between TTR Val30Met/non-TTR Val30Met mutations and early or late onset of disease.

### 3.2. Heart Transplantation

Heart transplantation can be feasible in patients with ATTR-amyloidosis and heart failure earlier than LTx, because the slowness of cardiac amyloid production and deposition. Combined heart and liver transplantation in younger patients has better prognosis than cardiac transplantation alone, probably because cardiac graft does not contain preformed amyloid deposits (nidus) to facilitate the addition of wild-type ATTR.

LTx has a positive impact on patient prognosis, but it is reserved to a restricted range of patients because of the procedural limitations and its uncertain benefits, except for Val30Met mutation patients [36].

### 3.3. Specific Pharmacological Treatment

#### 3.3.1. Inhibition of TTR Gene Expression

Patisiran (ONPATTRO™) is the first small-interfering RNA-based drug approved for the treatment of adults with ATTRv-related polyneuropathy both in USA and in European Union. It specifically acts on a genetically stored sequence in the untranslated 3′ region of the entire mutant and wild-type TTR mRNA. It is formulated in the form of lipid nanoparticles, administered through intravenous infusion, and it causes the catalytic degradation of TTR mRNA in the liver, resulting in reduced serum levels of the TTR protein. The efficacy and tolerability of Patisiran have been evaluated in a randomized double-blind trial, the APOLLO trial, in which patients were randomized 2:1 to receive Patisiran or placebo via intravenous infusion once every 3 weeks for 18 months. Patisiran-treated patients have been shown a better outcome at month 18, as demonstrated by decreased mean left ventricular wall thickness, decreased *N*-terminal prohormone of brain natriuretic peptide (NT-proBNP) levels, and by promoting a reverse remodelling, improving clinical and cardiac manifestations of ATTR. In a phase II trial, Patisiran has shown an effective reduction in both mutant and wild-type TTR levels in patients with ATTR—familial polyneuropathy [37].

Inotersen (TEGSEDI™) is an antisense oligonucleotide, approved by FDA and EMA for ATTRv-related polyneuropathy patients, which inhibits transthyretin production by binding the TTR mRNA. Its effectiveness has been assessed in a randomized, double-blind phase 3 trial, performed in 172 patients with stage 1 (ambulatory patients) or stage 2 (ambulatory patients with assistance) hereditary transthyretin amyloidosis with polyneuropathy. Inotersen was proven to reduce clinical manifestations of polyneuropathy and to improve the quality of life. Only 3% of patients underwent glomerulonephritis or thrombocytopenia

as major adverse events, with one death associated with grade 4 thrombocytopenia [32]. Patients who completed a phase III trial were enrolled in NEURO-TTR trial in which patients with ATTRv and polyneuropathy were randomized to Inotersen (300 mg weekly) or placebo. The Inotersen group showed a good drug tolerance and a slower neurological decline during the follow-up [38].

### 3.3.2. Tetramer Stabilization

Targeted therapies have focused on small molecules that are able to stabilize TTR-tetramer, preventing TTR amyloid fibril formation.

Tafamidis (VYNDAQEL®), an approved drug for the treatment of wild-type or hereditary ATTR in adult patients with cardiomyopathy and polyneuropathy, binds selectively to the tiroxine binding sites, stabilizing the tetramer and preventing the dissociation into monomers [39,40].

Its effectiveness has been proven in a randomized double-blind study, the ATTR-ACT trial, in which 441 patients with ATTR-amyloid cardiomyopathy were randomized to receive Tafamidis or placebo for 30 months [41]. A reduction in all-cause mortality rate was observed in Tafamidis group compared to the placebo (78 of 264 (29.5%) vs. 76 of 177 (42.9%); hazard ratio 0.70, 95% confidence interval (CI) 0.51–0.96) and a lower rate of hospitalizations for cardiovascular events. Most benefits regarding mortality were seen after 18 months of follow-up treatment. The drug was well-tolerated and no differences in adverse events were reported between placebo and Tafamidis groups [26].

### 3.3.3. Degradation and Reabsorption of Amyloid Fibres

New therapeutic approaches are aimed at targeting serum amyloid P (SAP), involved in amyloidosis pathogenesis (both AL and ATTR).

Miridesap is a small molecule that causes a rapid depletion of circulating SAP via hepatic clearance. A decrease of more than 90% of SAP circulating levels, in addition to a depletion of SAP within amyloid deposits in histological samples, was observed in seven patients with systemic amyloidosis after Miridesap-based treatment [42,43]. The treatment with anti-SAP antibodies after Miridesap infusion was shown to improve liver function and deplete amyloid tissue deposits in a phase I trial including patients with systemic amyloidosis. However, cardiac disease was not an inclusion criterion. Another trial showed no significant improvement or cardiac-adverse effect after Miridesap in patients with cardiac damage [44].

Anti-SAP antibody therapy is no longer evaluated. Considerable interest is focused on monoclonal antibodies directed towards the aggregates forms of ATTR to facilitate their uptake into macrophages. These antibodies show a rapid amyloid deposit removal and degradation of amyloid deposition with an improvement of the cardiac performance in a novel in vivo models of wtATTR [45].

Future research is focused on combinations of drugs with different mechanisms of action for a synergy approach, to prevent amyloid deposition and to improve clearance of amyloid deposits.

## 4. Conclusions

Many advances and novel opportunities in the treatment of CA have become available in recent years, and a disease with a poor prognosis has been revalued as a more manageable and possibly curable condition. The list of therapeutic options is rapidly expanding, with new options for CA targeting the multiple phases of amyloid cascade. Therefore, clinical competence and cardiologist's awareness are crucial to get an early diagnosis and to obtain a better prognosis.

**Author Contributions:** Conceptualization, F.I. and A.E.; methodology, M.C.; software, M.C.; validation, M.C., A.E. and F.I.; formal analysis, M.D.M.; investigation, A.I.; resources, R.P.; data curation, M.D.M.; writing—original draft preparation, F.I.; writing—review and editing, M.G.M.; visualization, M.D.M.; supervision, A.E. All authors have read and agreed to the published version of the manuscript.

**Funding:** This research received no external funding.

**Institutional Review Board Statement:** Not applicable.

**Informed Consent Statement:** Not applicable.

**Data Availability Statement:** Not applicable.

**Conflicts of Interest:** The authors declare no conflict of interest.

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
