# Peer review of "Cardiac Amyloidosis Therapy: A Systematic Review"

_cardiogenetics, doi:10.3390/cardiogenetics11010002_

Round 1

Reviewer 1 Report

The article has scientific value and it's well written.

I would suggest considering to increase the size type in the figures because it's a little hard to read all the contents. Also, get a higher quality for the figures because in the version I checked they seem blurred and not well defined. 

Author Response

The amendments are attached. Tanks

Reviewer 2 Report

In the manuscript, titled "Cardiac amyloidosis therapy: A systematic review", Iodice F et al., provides a comprehensive analyses of therapeutic modalities currently available for cardiac amyloidosis. There are only minor comments.

  1. Authors could rephrase the sections classical therapies targeting the plasma cells and therapeutic strategies targeting the plasma cells. Not clear what is the difference.
  2. Also, it would be better to number the sections for better organization and clarity.
  3. Section "Pharmacological Therapy" is also somewhat confusing. Authors should consider rephrasing the title.
  4. Supportive care section is somewhat appearing in the middle of the article. Is it related to any of the sections above or refers to general care during various therapies.
  5. The conclusion section could be expanded to discuss more about some the challenges or limitations of various treatment regimens and the future direction.

Author Response

The amendments are attached. Tanks
